# Brain Pathology in Mucopolysaccharidoses (MPS) Patients with Neurological Forms

**DOI:** 10.3390/jcm9020396

**Published:** 2020-02-01

**Authors:** Gustavo M. Viana, David A. Priestman, Frances M. Platt, Shaukat Khan, Shunji Tomatsu, Alexey V. Pshezhetsky

**Affiliations:** 1Division of Medical Genetics, CHU Ste-Justine Research Centre, Montreal, QC H3T 1C5, Canada; gvianabiomed@gmail.com; 2Department of Biochemistry, Federal University of São Paulo (UNIFESP), São Paulo 04044-020, SP, Brazil; 3Department of Pharmacology, University of Oxford, Oxford OX1 3SZ, UK; david.priestman@pharm.ox.ac.uk (D.A.P.); frances.platt@pharm.ox.ac.uk (F.M.P.); 4Nemours/Alfred I. duPont Hospital for Children, Wilmington, DE 19801, USA; Shaukat.Khan@nemours.org (S.K.); Shunji.Tomatsu@nemours.org (S.T.)

**Keywords:** mucopolysaccharidosis, brain pathology, neuroinflammation, glycosaminoglycans, glycosphingolipids, protein misfolding

## Abstract

Mucopolysaccharidoses (MPS) are the group of lysosomal storage disorders caused by deficiencies of enzymes involved in the stepwise degradation of glycosaminoglycans. To identify brain pathology common for neurological MPS, we conducted a comprehensive analysis of brain cortex tissues from post-mortem autopsy materials of eight patients affected with MPS I, II, IIIA, IIIC, and IIID, and age-matched controls. Frozen brain tissues were analyzed for the abundance of glycosaminoglycans (heparan, dermatan, and keratan sulfates) by LC-MS/MS, glycosphingolipids by normal phase HPLC, and presence of inflammatory cytokines interleukin-6 (IL-6) and tumor necrosis factor superfamily member 10 (TNFSF10) by Western blotting. Fixed tissues were stained for the markers for microgliosis, astrogliosis, misfolded proteins, impaired autophagy, and GM2 ganglioside. Our results demonstrate that increase of heparan sulfate, decrease of keratan sulfate, and storage of simple monosialogangliosides 2 and 3 (GM2 and GM3) as well as the neutral glycosphingolipid, LacCer, together with neuroinflammation and neuronal accumulation of misfolded proteins are the hallmarks of brain pathology in MPS patients. These biomarkers are similar to those reported in the corresponding mouse models, suggesting that the pathological mechanism is common for all neurological MPS in humans and mice.

## 1. Introduction

More than 30% of patients affected with lysosomal diseases have mucopolysaccharidoses (MPS), disorders affecting the enzymes involved in the stepwise degradation of glycosaminoglycans (GAGs) [1,2,3,4]. Seven MPS are caused by genetic deficiencies in enzymes participating in heparan sulfate (HS) catabolism. Three of them also have blocks in dermatan sulfate (DS) degradation, therefore resulting in accumulation of both DS and HS [5]: MPS I (Hurler–Scheie syndrome, α-iduronidase deficiency), MPS II (Hunter syndrome, iduronate sulfatase deficiency), and MPS VII (Sly syndrome, glucuronidase deficiency). Defects in the remaining four enzymes, α-N-acetylglucosaminidase, heparan *N*-sulfatase, HGSNAT, and *N*-acetylglucosamine 6-sulfatase, cause blocks only in the catabolism of HS. They are classified as variants of a single disorder, MPS III or Sanfilippo syndrome [5]. Other forms of MPS are associated with storage of DS and chondroitin sulfate, such as MPS VI, or keratan sulfate (KS), such as MPS IV, but neurological manifestations are caused only by accumulation of HS.

The variability in behavioral phenotypes for particular MPS subtypes is striking. The behavior of MPS I and MPS VII patients is generally normal and even the severely affected children are usually placid, calm, and sometimes overcautious, whereas MPS III patients show hyperactivity, aggressive behavior, insomnia, and autistic features such as lack of fear [5,6]. The neurological phenotype of MPS II is highly variable from severe in rapidly progressing cases to mild or even absent in slowly progressing cases [7]. No specific therapy is currently approved for neurological MPS, making them the biggest group of untreatable lysosomal diseases; however, numerous preclinical studies and clinical trials are being conducted [8].

Understanding the pathological basis underlying the central nervous system (CNS) manifestations in neurological MPS is crucial for developing novel therapies. Besides, it allows the identification of specific biomarkers of the disease both for monitoring disease progression and determining the efficacy of new therapies. Previous studies provided evidence that the accumulation of HS in neurological MPS triggers a multifaceted mechanism leading to neuronal malfunctioning and eventually to neurodegeneration. This mechanism involves general inflammation reactions, release of multifunctional cytokines and oxidative stress, pathological changes in the mitochondrial system and vesicular transport, progressive accumulation of lipid and protein aggregates, and closely resembles pathological cascades in age-related neurodegenerative tauopathies [9,10,11,12]. While these studies provide crucial information essential for our understanding of the diseases, it is important to understand that the majority have been conducted using the animal (mainly mouse) models of the disease, sometimes demonstrating clinical phenotypes different from those of the human patients. For example, all three mouse models of Sanfilippo disease (MPS IIIA, IIIB, and IIIC) develop urinary retention incompatible with their survival for longer than 8–11 months and not present in the human patients [13,14,15]. The alpha-*L*-iduronidase knockout MPS I mice show extensive dysostosis [16] as well as a progressive motor dysfunction [17]; their neurological phenotype is considerably milder as compared with that of rapidly progressing Hurler patients [18]. Several studies described analysis of CNS tissues of human MPS patients; however, the majority of them involved only one (and rarely more than two) type of MPS, preventing comparison of the brain pathology across diseases [19].

In the current study, we report the first comprehensive analysis of frozen and fixed brain cortex tissues from post-mortem autopsy materials of eight patients affected with MPS I, II, IIIA, IIIC, and IIID as well as age-matched controls. Our results demonstrate that an increase of HS (HS/DS), a decrease of KS, secondary lipidome changes, neuroinflammation, and protein folding defects are the most striking features of CNS pathology in MPS patients with neurological forms.

## 2. Patients and Methods

### 2.1. Patients

Ethical approval for the research involving human tissues was granted by Research Ethics Board (Comité d’éthique de la recherche numéro FWA00021692) of Centre hospitalier universitaire Sainte-Justine (CHU Ste.-Justine). The approval number is: 2020-2365. Frozen or fixed with PFA, cerebral tissues from MPS patients and age-matched controls were provided by National Institutes of Health (NIH) NeuroBioBank (project 1071, MPS Synapse) together with clinical descriptions and results of the neuropathological examination.

### 2.2. Analysis of Brain Glycosaminoglycans by Targeted LC-MS/MS

Disaccharides were produced from polymer GAGs by digestion with chondroitinase B, heparitinase, and keratanase II, resulting in DS (di-0S), HS (diHS-NS, diHS-0S), and KS (mono-sulfated KS, di-sulfated KS). The enzymes were obtained from Seikagaku Corporation (Tokyo, Japan), and chondrosine was used as an internal standard (IS). Unsaturated disaccharides, ΔDiHS-NS, 2-deoxy-2-sulfamino-4-*O*-(4-deoxy-a-*L*-threo-hex-4-enopyranosyluronicacid)-*D*-glucose; ΔDiHS-0S, 2-acetamido-2-deoxy-4-*O*-(4-deoxy-a-*L*threo-hex-4-enopyranosyluronicacid)-*D*-glucose; ΔDi-4S (DS), 2-acetamido-2-deoxy-4-*O*-(4-deoxy-a-*L*-threo-hex-4-enopyranosyluronic acid)-4-*O*-sulfo-*D*-glucose; mono-sulfated KS, Galβ1-4GlcNAc (6S) and di-sulfated KS, Gal (6S) β1-4GlcNAc(6S) were obtained from Seikagaku Corporation (Tokyo, Japan) and used to make standard curves. Stock solutions of ΔDiHS-NS (100 μg/mL), ΔDiHS-0S (100 μg/mL), ΔDi-4S (250 μg/mL), mono- and di-sulfated KS (1000 μg/mL), and IS (5 μg/mL) were prepared separately in Milli-Q water. Standard solutions of ΔDiHS-NS, ΔDiHS-0S, ΔDi-4S (7.8125, 15.625, 31.25, 62.5,125, 250, 500, and 1000 ng/mL), and mono- and di-sulfated KS (80, 160, 310, 630, 1250, 2500, 5000, and 10000 ng/mL) were each mixed with IS solution (5 μg/mL).

Frozen brain tissues (40–60 mg) were homogenized in 1.5 mL cold acetone for 10 sec using Polytron (Mount Holly, NJ). Homogenates were transferred to 2 mL Eppendorf tubes and kept at −80 °C for 20 min. Samples were centrifuged for 30 min at 12,000× *g* at 4 °C. The acetone was removed and pellets were dried in a vacuum centrifuge. The pellets were resuspended in 200 µL of 0.5 N NaOH and incubated at 50 °C for 2 h. Samples were neutralized with 100 µL of 1 N HCl to pH 7.0. Sodium chloride was added to a final concentration of 3M, followed by centrifugation at 10,000× *g* for 5 min at a room temperature (RT). The supernatants were transferred to new tubes, and 83.3 µL of 1 N HCl was added to make pH acidic (around 1.0). Then, tubes were centrifuged at 10,000× *g* for 5 min at RT. The supernatants were transferred to the new tubes and 83.3 µL of 1 N NaOH was added to increase pH to 7.0. The samples were diluted 2-fold with 1.3% potassium acetate in 100% ethanol and centrifuged at 12,000× *g* for 30 min at 4 °C. Supernatants were removed and pellets washed with 80% cold ethanol. Finally, the pellets were dried at RT, dissolved in 100 µL of 50 mM, Tris–HCl buffer (pH 7.0), and kept at −20 °C.

Ten microliters of each brain sample or standard and 90 μL of 50 mM Tris–HCl buffer (pH 7.0) were transferred to the wells of AcroPrep™ Advance 96-Well Filter Plates with Ultrafiltration Omega 10 K membrane filters (PALL Corporation, NY, USA). Then, 40 μL of the solution containing chondroitinase B (0.5 mU/sample), heparitinase and keratanase II (both, 1 mU/sample), and IS solution (5 μg/mL) followed by 60 μL of 50 mM Tris-hydrochloric acid buffer were also added to each well. The filter plate was placed on a 96-well plate, incubated at 37 °C overnight and centrifuged at 2500× *g* for 15 min.

The chromatographic system consisted of 1260 Infinity Degasser, binary pump, autoinjector, thermostatic column compartment, and 1290 Infinity Thermostat (Agilent Technologies, Palo Alto, CA, USA) and a Hypercarb column (2.1 mm internal diameter (i.d.) 50 mm, 5 μm, Fisher Scientific, Pittsburg, PA, USA) with hypercarb guard (2.1 mm i.d. 10 mm, 5 μm, Cole-Parmer, IL, USA). The column temperature was kept at 60 °C. The mobile phases were 100 mM ammonia (A) and 100% acetonitrile (B). The gradient conditions were programmed as follows: The initial composition of 100% A was held for 1 min, linearly modified to 30% B by 4 min, maintained at 30% B until 5.5 min, returned to 0% B by 6 min, and maintained at 0% B until 10 min. The flow rate was 0.7 mL/min. The 6460 Triple Quad mass spectrometer (Agilent Technologies) was operated in the negative ion detection mode with thermal gradient focusing electrospray ionization (Agilent Jet Stream technology, AJS). The parameters of jet stream technology were as follows: Drying gas temperature, 350 °C; drying gas flow, 11 L/min; nebulizer pressure, 58 psi; sheath gas temperature, 400 °C; sheath gas flow, 11 L/min; capillary voltage, 4000 V; and nozzle voltage, 2,000 V. Specific precursor and product ions, mass/charge (m/z), values were used to quantify each disaccharide: IS, 354.3, 193.1; DS, 378.3, 175.1; mono-sulfated KS, 462, 97; di-sulfated KS, 542, 462; diHS-NS, 416, 138; and diHS-0S, 378.3, 175.1. DS was measured as di-0S after the digestion of di-4S by a 4S-sulfatase present in the preparation of chondroitinase B. The injection volume was 5μL with a total run time of 10 min per sample. The peak areas for all components were integrated automatically using QQQ Quantitative Analysis software (Agilent Technologies), and peak area ratios (area of analyses/area of IS) were plotted against concentration by weighted linear regression. Raw data of LC-MS/MS were automatically preserved. The concentration of each disaccharide was calculated using QQQ Quantitative Analysis software.

### 2.3. Analysis of Brain Glycosphingolipids by HPLC 

Glycosphingolipids (GSLs) were isolated and analyzed essentially as described [20]. Brain tissues were homogenized in CHU Ste.-Justine in water using an Ultra-Turrax T25 probe homogenizer (IKA, Germany) and sent to Oxford for the analysis of brain glycosphingolipids. Protein concentrations in the homogenates were determined using bicinchoninic acid (BCA) assay (Sigma, UK). Lipids from the homogenates were then extracted with chloroform/methanol, and GSLs were further purified using solid-phase C18 columns (Telos, Kinesis, UK). Eluted lipid fraction GSLs were dried under nitrogen and digested overnight with recombinant endoglycoceramidase I, kindly supplied by Orphazyme. The released glycans were then fluorescently labelled with anthranilic acid (2AA) prior to purification using DPA-6S SPE (Solid Phase Extraction) amide columns (Supelco, PA, USA). Purified, 2AA-labelled glycans were then separated and quantified by normal-phase high-performance liquid chromatography (NP-HPLC) as described previously [20]. In order to calculate GSL molar quantities, 2.5 pmol of a 2AA-labelled chitotriose standard (Ludger, UK) was also included in the HPLC sample group.

### 2.4. Immunofluorescence and Confocal Microscopy

Formalin-fixed human brains were embedded in optimum cutting temperature compound (OCT), cut into 40-µm-thick sections and stored at −80 °C in cryopreservation buffer (0.05 M sodium phosphate buffer, pH 7.4, 15% sucrose, 40% ethylene glycol). For immunofluorescence analysis, brain slices were washed with phosphate-buffered saline (PBS), pH 7.4, and blocked with 5% bovine serum albumin with 0.3% Triton X-100 for 1 h. Slices were then incubated with the following primary antibodies: Rabbit anti-glial fibrillary acidic protein or GFAP/8-1E7 (1:200, 8-1E7, deposited to the Developmental Studies Hybridoma Bank by De Blas, Angel L.; information available at https://dshb.biology.uiowa.edu/8-1E7), rabbit anti-LC3B (1:200, GeneTex, Irvine, CA, USA; catalog number: GTX127375), and mouse humanized anti-GM2/KM966 (1:100, kindly provided by Dr. Nobuo Hanai, Dr. Akiko Furuya, and Kyowa Hakko Kirin Co., Ltd.). Secondary antibodies (Life Technologies, USA) were Alexa Fluor^®^ 488 donkey anti-rabbit IgG (catalog number: R37118) and Alexa Fluor^®^ 633 goat anti-mouse IgG (catalog number: A-21050). For microglia visualization, samples were labeled in situ (1:20) with isolectin B4 from *Griffonia simplicifolia* (GS-ILB4), AlexaFluor^TM^ 568 conjugate (ILB4, Life Technologies, USA, catalog number: I21412). For amyloid aggregates, brain slices were washed with PBS, incubated with 0.3% Triton X-100 in PBS for 1 h at RT and stained with 0.05% thioflavin S (Sigma, USA, catalog number: T1892) in PBS in the dark for 15 min, washed in 50% ethanol twice for 1 min and, then, in water for 3 min. The slices were then mounted using Prolong Gold Antifade Mounting Medium with 4′,6-diamidino-2-phenylindole (DAPI;Life Technologies, USA, catalog number: P36931). Representative images from the whole tissue were captured with a Leica TCS SP8 Confocal Laser Scanning Microscope (Leica Microsystems, DE). Fluorescence observations were made through oil-immersion Plan-Apochromat 63x objectives (numerical aperture 1.4). Images were represented as maximum intensity projections corresponding to the z-series of all confocal stacks. Images were quantified using the ImageJ software (US National Institutes of Health, USA) and mounted and processed using Photoshop (Adobe Inc., USA).

### 2.5. Western Blots

Brain tissues (frontal part of a hemisphere, approximately 25% of the brain) were homogenized in radioimmunoprecipitation assay (RIPA) buffer (50 mM Tris-HCl, pH 7.4, 150 mM NaCl, 1% NP−40, 0.25% sodium deoxycholate, 0.1% SDS, 2 mM EDTA, 1 mM phenylmethylsulfonyl fluoride (PMSF), Roche protease and phosphate inhibitor cocktails, 2.5 mL per 1 g of tissue). The homogenates were kept on ice for 30 min and centrifuged at 13,000 RPM at 4 °C for 25 min. The supernatant was collected and centrifuged again for 15 min. Total protein concentration was measured using the Pierce BCA Protein Assay Kit (Thermo Fisher Scientific) and equalized. Protein extracts (50 μg) were resolved by SDS–PAGE on 10%–15% gels (29:1 acrylamide/bis-acrylamide) and transferred to nitrocellulose membranes (Millipore, USA). Membranes were blocked with 5% BSA in TBS-T (Tris-buffered saline with 0.1% Tween-20, pH 8.4) for 1 h at room temperature and then incubated overnight at 4 °C with the following primary antibodies (diluted 1:1000 in TBS-T/1% BSA): rabbit anti-AFFN-TNFSF10-11F2 (tumor necrosis factor superfamily member 10 or TNFSF10, deposited to the Developmental Studies Hybridoma Bank by EU Program Affinomics, catalog number: AFFN-TNFSF10-11F2, RRID:AB_2617959) and rabbit anti-CPTC-IL6-1 (IL-6; deposited to the DSHB by Clinical Proteomics Technologies for Cancer, catalog number: CPTC-IL6-1, RRID:AB_2617278). After several washings in TBS–T, membranes were incubated with horseradish peroxidase-conjugated anti-mouse IgG (1:2500, Cell Signaling Technologies, USA, catalog number: 7076), anti-rabbit IgG (1:2,500, Cell Signaling Technologies, USA, catalog number: 7074) or rabbit anti-goat IgG (1:10,000, Merck, USA, catalog number: A-8919). The immunoblots were revealed with Pierce enhanced chemiluminescence (ECL) Western blotting chemiluminescence substrate (Thermo Fisher Scientific, USA). Bands were quantified using ImageJ software (US National Institutes of Health, USA). Total GAPDH protein was used for data normalization.

## 3. Results

### 3.1. Patients

Frozen and/or paraformaldehyde (PFA)-fixed somatosensory cortex tissues from eight MPS patients (one MPS I, two MPS II, two MPS IIIA, one MPS IIIC, and two MPS IIID) and seven non-MPS, controls matched for age and sex, collected at post-mortem autopsy were obtained from National Institutes of Health (NIH) NeuroBioBank (project 1071, MPS Synapse). The age, the cause of death, sex, race, and available clinical and neuropathological information for the patients and controls are shown in Table 1. All MPS patients had complications from their primary disease and died in the first-third decades of life except MPS II patient 902, who was suffering from a non-neurological form of the disease and died at 42 years of age from MPS II-related pneumonia. None of the patients had received enzyme replacement therapy (ERT) or hematopoietic stem cell transplantation (HSCT). Among non-MPS controls, two died of accident-related injuries, two of asthma complications, two of atherosclerotic cardiovascular disease, and one of smoke inhalation (Table 1).

### 3.2. Changes in GAG Profiles

To study the impact of the deficiencies of the enzymes involved in catabolism of GAGs, we analyzed their levels in brain cortex tissues. Tandem mass spectrometry (LC-MS/MS) was used to determine the concentration of disaccharides derived from the three GAGs known to accumulate in MPS diseases: ΔDi-0S (DS), ΔDiHS-NS, and ΔDiHS-0S (HS), as well as mono-sulfated and di-sulfated KS. Disaccharides were produced by specific enzyme digestion of each GAG, and quantified by negative-ion mode of multiple reaction monitoring. Analyses of the data demonstrated that the GAG profiles in control brains were significantly different from those of the patients with neurological MPS (Figure 1A). In control brain tissues, mono-sulfated KS was the most abundant, followed by ΔDi-0S, ΔDiHS-0S, and ΔDiHS-NS. In the cortex tissues of all MPS patients, we observed increased levels of ΔDiHS-NS (2.15 median value in the patients vs 0.84 in controls, *p* = 0.0012 in Mann–Whitney test) and ΔDiHS-0S (7.0, patients, vs. 1.38, controls, *p* = 0.0003). At the same time, the levels of mono-sulfated and di-sulfated KS were reduced in all MPS patients (mono-sulfated KS, 1.15, patients, vs. 6.70, controls, *p* = 0.0003; di-sulfated KS, 0.069, patients, vs. 0.144, controls, *p* = 0.0003).

Analyses of the GAG levels in individual patients (Figure 1B) confirmed that DS increased only in the MPS I and MPS II, but not in MPS III patients. Levels of DS in the two MPS II patients were significantly different; the non-neurological MPS II patient 902 had ~7-fold lower levels (marked with an arrowhead in Figure 1A,B) of DS than the MPS II patient HBCB1801OC who died at the age of 13 years, suffering from a more severe MPS II form with a CNS involvement. All patients, except the non-neurological MPS II patient 902, showed significantly increased levels of brain HS and decreased levels of KS.

### 3.3. Alteration of Glycosphingolipid Profiles

The glycan chains of total glycosphingolipids extracted from the brain tissues were fluorescently labelled with anthranilic acid and analyzed by normal-phase HPLC. Quantification of chromatograms demonstrated that brain glycosphingolipid composition was significantly altered in the MPS patients. The control brain samples were dominated by four ganglioside species: GM1a (Monosialoganglioside) (35.2 ± 2.1%), GD1a (Disialoganglioside) (26.9 ± 2.3%), GD1b (13.9 ± 1.2%), and GT1b (Trisialoganglioside) (9.4 ± 0.9%). They were present at comparable levels, together representing ~85% of brain glycosphingolipids (Figure 2A). This correlated well with previously published rodent data [21]. Besides these four major species, we were able to identify 13 minor Glycosphingolipid (GSL) species of which only 4 exceed 1% of total glycolipids: LacCer (3.5 ± 1.2%), GM2 (2.3 ± 0.1%), α-2,6-SpGb and GD3 (both ~1.7%) and GM3 (1.05 ± 0.6%).

The glycosphingolipid composition was greatly altered in the brains of the MPS patients. The relative levels of three of four major complex gangliosides were significantly reduced. GM1a is reduced from 35.2 ± 2.1 to 25.1 ± 1.4% (*p* = 0.0012 in multiple comparison t-test), GD1a, from 26.9 ± 2.3 to 16.7 ± 1.3% (*p* = 0.0015), and GT1b from 9.4 ± 0.9% to 5.9 ± 0.8 (*p* = 0.015). GD1b also showed a trend for decrease, but it was not statistically significant. In contrast, simple gangliosides GM2, GM3, and LacCer became major components in the MPS brain samples (GM2 7.4 ± 1.081, 3.2-fold increase, *p* = 0.0006; GM3 10.3 ± 1.5, 10-fold increase, *p* = 5.2e^−005^; LacCer, 10.3 ± 1.6, 3-fold increase, *p* = 0.006).

In addition, several glycolipids showed significant increases but still remained to be relatively minor components of the cortical tissue (GD3 increased from 1.75 to 3.7%, *p* = 2.5e^−005^; Gb3, from 0.16 to 0.72%, *p* = 0.0003; GA1, from 0.69 to 1.51%, *p* = 0.007; and GA2, from 0.27 to 1.22%, *p* = 0.017). Importantly, the glycosphingolipid composition of the brain tissues of non-neurological MPS II patient 902 resembled more those of controls, than of MPS patients (GM1a 32.5%, GD1a 19.6%, GD1b 15.7%, and GT1b 9.1%). His tissues also showed slight elevation of LacCer (6%) and GM3 (2.8%) but not GM2 ganglioside (2.86%, marked with arrowheads on Figure 2B). When absolute values of simple GM2 and GM3 gangliosides were grouped according to the type of MPS disorder (Figure 2C), they showed remarkable resemblance to those of HS with higher levels in MPS I and, somewhat, lower levels in MPS II.

Accumulation of GM2 ganglioside in brain tissues was further confirmed by immunohistochemistry using the human–mouse chimeric monoclonal antibody, KM966 [22]. Numerous cells resembling by shape the pyramidal neurons and showing intensive staining by the antibody were present in the deep cortex layers of all MPS patients while being almost undetectable in the brain tissues of controls (Figure 2D).

### 3.4. Microastroglyosis and Neuroinflammation

The number of Glial fibrillary acidic protein (GFAP)-positive astrocytes and ILB4-positive microglia cells was augmented in all studied human MPS cortex tissues except for those of the non-neurological MPS II patient 902 (Figure 3A), suggesting that neuroinflammation previously reported in all mouse models was also heavily present in human patients. To test this further, we measured the protein levels of the inflammatory cytokines, IL-6, previously shown to be increased in patients and animal models across several MPS disorders [23,24,25]. We have also measured the levels of TNFSF10 (TRAIL), which has been associated with neuroinflammation in adult neurodegenerative diseases including Alzheimer’s disease and multiple sclerosis [26,27]. We found that in the majority of MPS samples, levels of IL-6 and TRAIL proteins showed a trend for an increase. Levels of both IL-6 and TRAIL were significantly increased when the group of all MPS patients was compared with the group of controls (*p* = 0.04 and *p* = 0.006 in Mann–Whitney test, respectively).

### 3.5. Autophagy Block and Neuronal Accumulation of Misfolded Proteins 

Impaired autophagy is a characteristic feature of cells in many lysosomal disorders [28]. It has been specifically shown for the neurons of the mouse models of neurological MPS disorders and proposed to be responsible for storage of misfolded protein aggregates, with accumulation of damaged nonfunctional mitochondria [15,29]. To test whether this is also the case for brain tissues from human MPS patients, we analyzed the distribution of light chain 3B protein (LC3B) in cortical neurons (Figure 4A). LC3B not detectable in the neurons of control brains has been localized to the cytoplasmic puncta in MPS neurons characteristic for the cells with impaired fusion of autophagosomes with lysosomes and increased number of autophagosomes containing secondary storage materials.

Neurons in the MPS brains showed increased fluorescence after staining with thioflavin S (Figure 4B). This dye binds to beta sheet-rich structures and displays enhanced fluorescence, which is useful for detection of misfolded proteins and specifically amyloid aggregates in the brains of Alzheimer patients. The increased thioflavin staining is, therefore, suggestive of amyloid accumulation in the cortex of MPS patients, previously detected in the medial entorhinal cortex neurons of the mouse models of MPS I, IIIB, and IIIC.

## 4. Discussion

Our report identifies the neuronal accumulation of GAGs, GM2 and GM3 gangliosides, and misfolded proteins together with neuroinflammation as the hallmarks of CNS pathology in the neurological MPS disorders. All patients accumulated HS as detected by the increased levels of disaccharides *N*- or *O*-sulfated at the nonreducing glucosamine residue. The MPS I and MPS II patients also showed a remarkable increase in DS because it was degraded by α-L-iduronidase and iduronate sulfatase which are, respectively, affected in these diseases. All patients also demonstrated a significant ~5-fold reduction of mono-sulfated and di-sulfated KS. This result was completely unexpected since previous work showed increased KS levels in both blood plasma and urine of MPS I, II, III, VI, and VII patients [30,31]. Similarly, increased KS levels were also detected in blood from the mouse models of MPS I, MPS IIIA, and MPS VII and thought to be caused by secondary inhibition of the KS-degrading enzyme, *N*-acetyl-galactose amine 6-sulfate sulfatase (GALNS), by the primary storage product, HS [32]. Other explanations for the secondary elevation of blood KS were also proposed, including the induction of KS synthesis by accumulated GAGs and/or inflammation, and its secretion into the circulation from damaged cartilage [33]. Importantly, the levels of blood KS correlate with the degree of skeletal dysplasia in several mouse MPS models, suggesting that secondary KS elevation may be responsible for the bone phenotype [32]. It is possible to speculate that the discrepancy in the direction of KS changes in the blood and the brain tissue can be related to the CNS damage and release of KS from destroyed brain cells into the blood or to the secondary inhibition of KS proteoglycan synthesis. Further studies are, therefore, necessary to identify both the mechanism underlying the strong KS reduction in the brain cortex of MPS patients and the potential pathological consequences of these changes.

Our results demonstrate the drastic alteration of GSL composition in patients with neurological forms of MPS I, II, IIIA, IIIC, and IIID. All these patients show a relative reduction of complex gangliosides (GM1a, GD1a, and GT1b) and an increase in simple (GM2, GM3) gangliosides as well as in neutral GSLs (Gb3, GA1, GA2, and LacCer). In contrast, in the patient with a nonneurological form of MPS II the sphingolipid composition was similar to that of normal controls. The increased levels of GM2 and GM3 gangliosides in the brain tissues were consistent with previously reported storage of these lipids in the mouse models of MPS IIIA, B, and C [13,14,15,28], but to our knowledge the marked increase of the neutral asialosphingolipids in the brains of MPS patients has not been reported previously.

The mechanism behind the secondary changes in brain sphingolipids is still discussed in the literature, and a majority of authors tend to agree that the accumulation of gangliosides in lysosomal storage disorders is related to the secondary impairment of autophagy/catabolism of autophagosomal content and lysosomal function. Direct evidence for the latter is still missing. It has been proposed to result from either the direct inhibition of lysosomal enzymes by accumulating substrates (e.g., GAGs) or from general disruption of the endolysosomal environment crucial for sphingolipid degradation (pH change, lack of saposin-substrate interaction, etc.) [34]. Others have suggested that gangliosides accumulating in MPS are of nonlysosomal origin and are induced by altered Golgi function [35] or by distorted vesicular trafficking of lipids causing the build-up of exogenous LacCer in late endosomes and lysosomes [36,37]. Our data indicated that the changes were very specific for individual sphingolipids and consistent across multiple MPS, suggesting the involvement of common and precise biochemical mechanisms.

Neuroinflammation, thought to be triggered by HS storage affecting toll-like receptors (TLR) of microglia, was previously reported for all neurological MPS patients and mouse models (see, for example, [38]). It is recognized as an important component of the pathophysiological mechanism of the disease, although the work of Ausseil et al. demonstrated that, in TLR4-knockout MPS IIIB mice, neurodegeneration can occur independently of microglial activation suggesting the existence of alternative pathways [39]. Our current data demonstrated high levels of astromicroglyosis in the cortex of all MPS patients suggesting elevated neuroinflammation. Unfortunately, mRNA species were not entirely intact in all tissues studied, which prevented us from conducting RT-Q-PCR-based analysis of cytokines as we have done previously for the MPS IIIC mice [15]. However, Western blotting demonstrated increased levels of inflammatory cytokines, IL-6 and TNFSF10 (TRAIL), previously associated with neuroinflammation in humans. Interestingly, IL-6 levels were not increased in the mouse models of MPS I, IIIA, and IIIB [38].

Finally, increased thioflavin S staining suggested that cortical neurons of all MPS patients studied accumulated misfolded protein aggregates. While it is tempting to speculate that they can contain misfolded amyloid, this still needs to be experimentally verified. Although the degree of staining varied between the individual samples, the MPS patients showed the trend for an increase as compared with controls of the same age. Our previous data defined accumulation of densely packed material displaying a strong autofluorescence as a determining feature of brain neurons in MPS IIIC mice at the advanced stage of the disease [15]. These granules were positively stained with antibodies against the misfolded form of the subunit C of mitochondrial adenosine triphosphate synthase (SCMAS), and their ultrastructural pattern strongly resembled that in neuronal ceroid lipofuscinoses suggesting that they are derived from autophagosomes and linking them to impaired autophagy. Our present data showed that all MPS patients and none of the controls showed the hallmark of the autophagy block, cytoplasmic LC3B-positive puncta.

Previously, we detected the specific sequence and pattern for the appearance of biomarkers of brain pathology in MPS IIIC mouse model: First GAGs in microglia, then neuroinflammation and neuronal ganglioside storage, followed by ceroid materials and neurodegeneration [15]. Based on this, we hypothesized that the disease starts with the accumulation of primary storage materials that induce general inflammation reactions in the brain, eventually leading to neuronal death [15]. Our current study identified the same biomarkers in post-mortem autopsy cortex samples of the human MPS I, II, IIIA, IIIC, and IIID patients suggesting that the pathological mechanism is common for all neurological MPS in humans and mice.

## Figures and Tables

**Figure 1 jcm-09-00396-f001:**
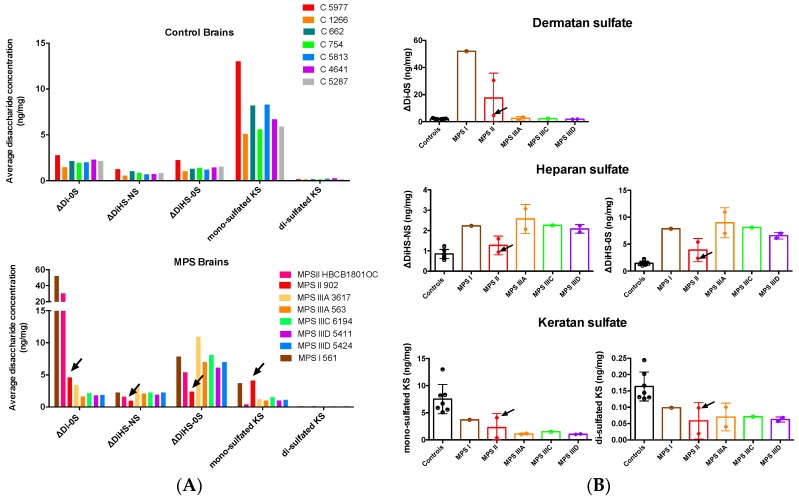
Glycosaminoglycans (GAGs) levels in the brains of mucopolysaccharidosis patients and controls. Levels of disaccharides produced by enzymatic digestion of dermatan sulfate (DS) (ΔDi-0S, ΔDiHS-NS) and heparan sulfate (HS) (ΔDiHS-0S) as well as mono-sulfated and di-sulfated keratan sulfate (KS) were measured by tandem mass spectrometry and expressed (**A**) for the individual MPS patients and controls or (**B**) grouped according to the type of MPS disorder. Statistical significance of changes was estimated using Mann–Whitney multiple comparison test. Arrows indicate values detected in the tissues of non-neurological MPS II patient 902.

**Figure 2 jcm-09-00396-f002:**
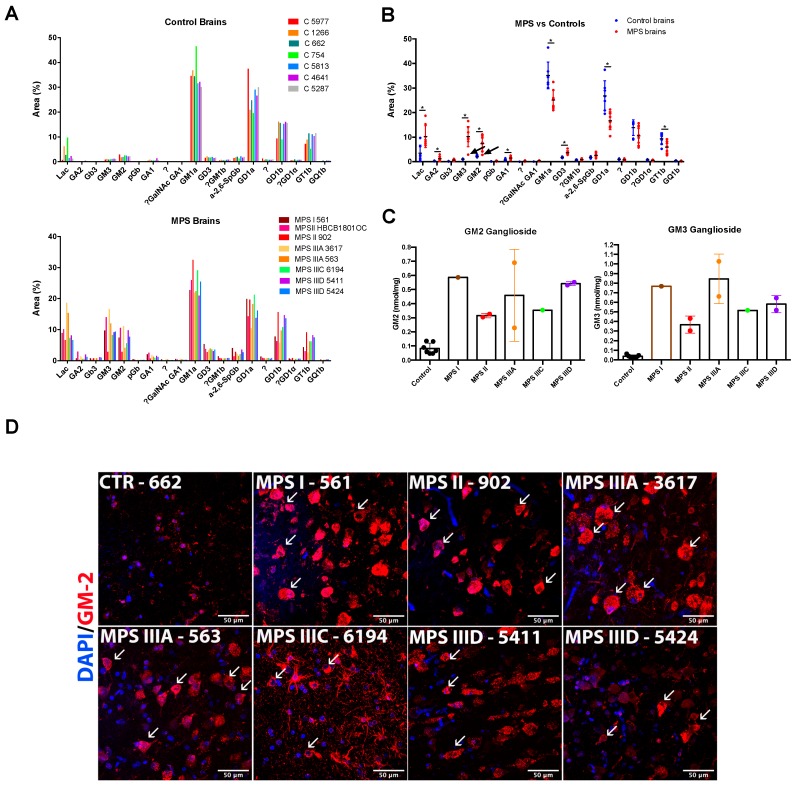
Sphingolipid levels in the brains of MPS patients and controls. Levels of glycans produced by enzymatic cleavage of total sphingolipid extracts of MPS patients and controls were measured by normal HPLC and plotted for the individual MPS patients and controls (**A**) or grouped for all patients and controls (**B**). The values show percentage of the specific lipid. The “?” means that the identity of the sphingolipid was unknown, or could not be confirmed. Statistical significance of changes was estimated using Mann–Whitney multiple comparison test (* *p* < 0.05). (**C**) Levels of simple GM2 and GM3 gangliosides grouped according to the type of MPS disorder. (**D**) Confocal microscopy images of brain cortex tissues of MPS I (561), MPS II (902), MPS IIIA (3617 and 563), MPS IIIC (6194), and MPS IIID (5411 and 5424) patients and a representative control (662) stained with antibodies against GM2 (red). The fixed brain tissues of the MPS II patient HBCB1801OC were not sufficiently preserved to perform cryosectioning and conduct immunofluorescent analysis. 4′,6-Diamidino-2-phenylindole (DAPI, blue) was used as the nuclear counterstain. Neurons with GM2 storage are shown with arrowheads. Scale bar: 50 µm.

**Figure 3 jcm-09-00396-f003:**
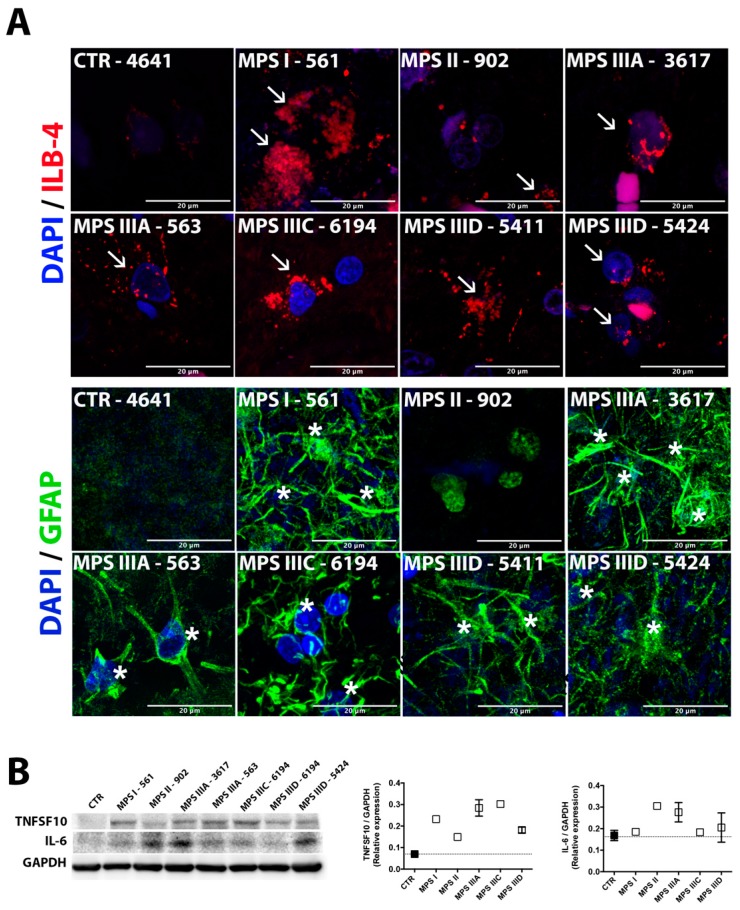
Astromicrogliosis in brain cortex tissues of human MPS patients is indicative of neuroinflammation. (**A**) Confocal microscopy images of brain cortex tissues of MPS I (561), MPS II (902), MPSIIIA (3617 and 563), MPS IIIC (6194), and MPS IIID (5411 and 5424) patients and a representative control (4641) stained with isolectin beta-4 (ILB-4,red) and antibodies against GFAP (green), markers for activated microglia and astrocytes, respectively. The fixed brain tissues of the MPS II patient HBCB1801OC were not sufficiently preserved to perform cryosectioning and conduct immunofluorescent analysis. DAPI (blue) was used as the nuclear counterstain. Activated microglia are marked with arrowheads and astrocytes are marked with asterisks; scale bar: 20 µm. (**B**) Western blotting analysis showing increased protein expression of the proinflammatory cytokines (tumor necrosis factor superfamily member 10 (TNFSF10) and interleukin 6 (IL-6) in the brain cortex protein extracts from MPS patients and combined controls (662, 754, 1266, 4641, 5287, 5813, and 5977). Glyceraldehyde 3-phosphate dehydrogenase (GAPDH) was used as a loading control. Data are expressed as the mean ± s.e.m.

**Figure 4 jcm-09-00396-f004:**
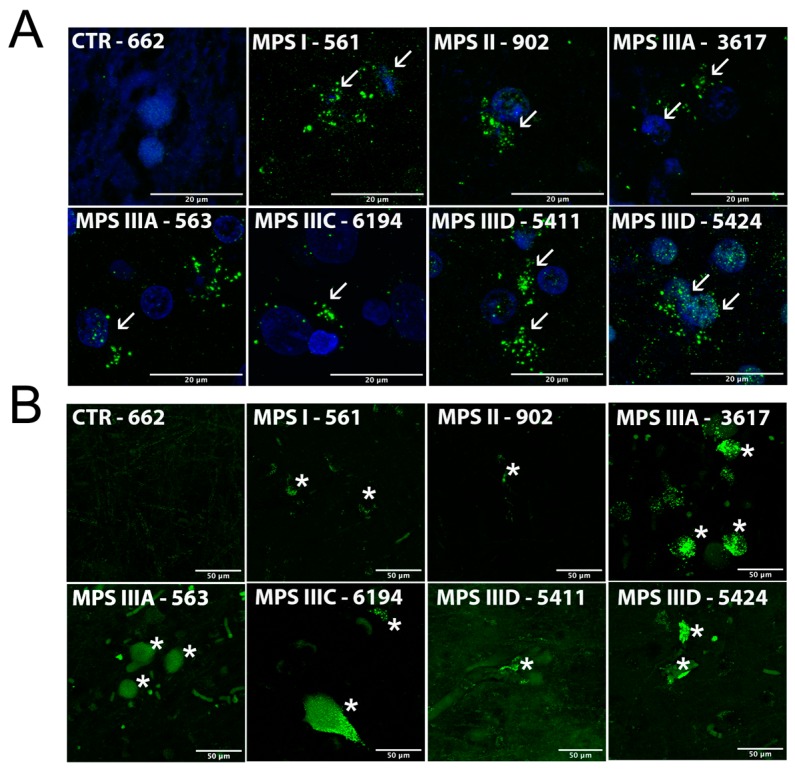
Impaired autophagic flux and accumulation of misfolded proteins in brain cortex tissues of human MPS patients. Representative confocal microscopy images showing (**A**) accumulation of light chain 3B protein (LC3B)-positive puncta (green) and (**B**) thioflavin S-positive protein aggregates (green) in brain cortex tissues from MPS I (561), MPS II (902), MPSIIIA (3617 and 563), MPS IIIC (6194), and MPS IIID (5411 and 5424) patients and a representative control (662). The fixed brain tissues of the MPS II patient HBCB1801OC were not sufficiently preserved to perform cryosectioning and conduct immunofluorescent analysis. DAPI (blue) was used as the nuclear counterstain. Neurons with LC3B puncta are marked with arrowheads and those stained with thioflavin S are markedwith asterisks. Scale bars: 20 µm (**A**) and 50 µm (**B**).

**Table 1 jcm-09-00396-t001:** Mucopolysaccharidoses (MPS) patients and control subjects used in the study.

UMBN	GUID	Disorder	Cause of Death	Age: Years, Days	Sex	Race	Clinical Information	Neuropathologic Findings
561	NDAR_INVYV182KRN	MPS I, Hurler Syndrome	Complications of disorder	6, 265	Female	Caucasian	Was oxygen dependent; had hydrocephalus, cardiomyopathy, chronic sinusitis, ear infections, blindness, hearing impairment, numerous pneumonias and hernias.	Neocortex with distended, "ballooned" neurons with mucin- and Alcian-blue positive material, also present throughout the central nervous system; occasional perivascular macrophages with similar material in the white matter; EM shows typical "zebra bodies" seen in mucopolysaccharidoses.
902	NDAR_INVTG497HU7	MPS II, Hunter Syndrome	Complications of disorder	42, 134	Male	Caucasian	Had multiple complications related to Hunter’s syndrome, including tracheobronchial malacia, recurrent bronchitis and pneumonia, had multiple repairs of anterior abdominal wall hernia, was blind and somewhat deaf, had bilateral carpal tunnel release, history of mitral and aortic insufficiency, congestive heart failure.	Gliosis with axonal degeneration, optic nerves bilaterally, with neuronal loss and gliosis, lateral geniculate nucleus, old hemorrhagic cystic infarct, right occipital cortex and white matter, periventricular benign epidermal cyst, right occipital.
3617	NDAR_INVFP950EUM	MPS IIIA, Sanfilippo A Syndrome	Complications of disorder	12, 38	Female	Caucasian	N/A	Neurons throughout the brain have enlarged cell bodies with foamy cytoplasm, mild gliosis and status spongiosis in adjacent parenchyma, these neuronal changes are particularly severe in cerebral cortex, Purkinje cell layer of the cerebellum and substantia nigra. The choroid plexus epithelial cells are similarly affected with slightly enlarged and vacuolated cytoplasm. The centrum semi-ovale is mildly gliotic, its perivascular spaces dilated, fibrotic and contain glitter cells and occasional lymphocytes.
563	NDAR_INVRR063YHC	MPS IIIA, Sanfilippo A Syndrome	Complications of disorder	11, 101	Female	Caucasian	Two years before the death was attending school although having trouble walking and eating, by the time of death was non-verbal with deteriorating psychomotor skills, self-injurious behavior, had problems with sleep and agitation, suffered from mitral valve prolapse with myxomatous changes and mild regurgitation.	N/A
6194		MPS IIIC, Sanfilippo C Syndrome	Acute pneumonia as a consequence of disorder	20, 95	Male	African-American	Had a history of developmental delays, Nissen fundoplication and G-tube, asthma, seizures, sleep problems, agitation, used hearing aids.	The brain showed cerebral atrophy, mild hydrocephalus, neuronal enlargement with positive cytoplasmic PAS, Alcian blue and LFB, perivascular cuffing of foamy macrophages, white matter vacuolation.
5411	NDAR_INVAB442TCG	MPS IIID, Sanfilippo D Syndrome	Complications of disorder	24, 280	Female	Caucasian	One of two siblings suffering from Sanfilippo D, had a progressive neurologic decline with loss of vision, verbal expression, continence; had rare seizures in spite of anticonvulsive treatment and was wheelchair bound for the last year of life.	Generalized cerebral atrophy and neuronal storage disorder.
5424	NDAR_INVUC095YP2	MPS IIID, Sanfilippo D Syndrome	Complications of disorder	23, 149	Female	Caucasian	One of two siblings suffering from Sanfilippo D, had a progressive decline in hearing, verbal and visual abilities, had no specific cardiopulmonary symptomatology, no seizures, wheelchair bound for the last two years of life.	Generalized cerebral atrophy and neuronal storage disorder.
HBCB_18_01_OC		MPS II, Hunter Syndrome	Complications of disorder	13, 0	Male	Caucasian	Had a history of developmental delays, scoliosis, asthma, tracheomalacia, GERD, seizures and mild aortic regurgitation.	Diffuse neuronal loss, gliosis in the cortex; abundant swollen neurons, greatest in the parietal and occipital cortex; severe neuronal loss and gliosis in the thalamus; depletion of processes and varying gliosis in the cerebellum.
662	NDAR_INVCK582GNX	Unaffected Control	Accident, multiple injuries	12, 356	Female	Caucasian		N/A
754	NDAR_INVJV820CBR	Unaffected Control	Asthma	11, 201	Female	Native Hawaiian/Pacific Islander		N/A
1266	NDAR_INVCX672EJ2	Unaffected Control	Arteriosclerotic cardiovascular disease	42, 0	Male	Caucasian		N/A
4641	NDAR_INVNG087HR2	Unaffected Control	Acute asthma	24, 288	Female	African-American		N/A
5287	NDAR_INVUB832RTY	Unaffected Control	Multiple injuries	23, 195	Female	Caucasian		N/A
5813	NDAR_INVWA136XNT	Unaffected Control	Atherosclerotic cardiovascular disease	20, 362	Male	African-American		N/A
5977	NDAR_INVAX199AGW	Unaffected Control	Smoke inhalation	6, 248	Female	Caucasian		N/A

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
