# Peer review of "Brain Pathology in Mucopolysaccharidoses (MPS) Patients with Neurological Forms"

_jcm, 2020, doi:10.3390/jcm9020396_

Round 1
Reviewer 1 Report
The present study is well structured and argued and certainly relevant for the whole scientific community working on MPSs. However, I suggest the following minor revisions to the authors:
Update the reference concerning the percentage of MPS patients (line 35). Implement the background regarding the behaviour of MPS II patients since it is one of the MPSs analysed in this study. The statement in lines 64-65 is not properly correct, as both the mentioned paper and Baldo et al 2012 (Behav Brain Res. 2012 Jul 15;233(1):169-75) show some defects in behavior, even if starting from 4 months. Please edit the sentence accordingly. Please add references in line 67. In line 76, and in almost all the manuscript, authors report the use of 8 samples, while in the abstract (line 18) they report the use of 10 samples. Please correct. Line 80: Table 1 is mentioned but is missing in the manuscript and it would be very useful for a better understanding of the analysed samples! In Figure 1A, in the graphic of the control brains, please change the range of the Y-axis from 0-20 to 0-15 to allow a more direct comparison with the MPS brains chart. In Figure 2C, 3 and 4 the MPSII HBCB1801OC sample is missing! Please add all the data also for it or specify in the text why it is not included! In Figure 2C, add a chart for GM2 and GM3 simple gangliosides, like the graph in Figure 1B. This will allow more direct visualization of the data for single MPS. In line 185-186, review [21] is not the proper one since it doesn’t really review LSDs with impaired autophagy but the mechanisms. In addition, several studies on LSDs and autophagy have been published in recent years. Therefore replace/add a newer and more appropriate review like Seranova et al 2017 (Essays Biochem. 2017 Dec 12;61(6):733-749). In Figure 4A, the scale bar is wrong (20 µm instead of 50 µm). In line 245 the reference concerning MPS IIIC is missing. In the “methods” section please specify the code number of the antibodies used where possible.
Author Response
The point-to-point response to the reviewer’s comments
Reviewer 1
The present study is well structured and argued and certainly relevant for the whole scientific community working on MPSs. However, I suggest the following minor revisions to the authors:
Update the reference concerning the percentage of MPS patients (line 35).
Response: We have added more recent references.
Implement the background regarding the behaviour of MPS II patients since it is one of the MPSs analysed in this study.
Response: We have expanded the description of the neurological MPS phenotypes to the MPS II and added a new reference (page 2, first paragraph, lines 47-49).
The statement in lines 64-65 is not properly correct, as both the mentioned paper and Baldo et al 2012 (Behav Brain Res. 2012 Jul 15;233(1):169-75) show some defects in behavior, even if starting from 4 months. Please edit the sentence accordingly.
Response: We thank the reviewer for pointing out this mistake. We have edited the sentence and added the reference to the work of Baldo et al. (page 2, second paragraph, lines 66-68).
Please add references in line 67.
Response: The references have been added.
In line 76, and in almost all the manuscript, authors report the use of 8 samples, while in the abstract (line 18) they report the use of 10 samples. Please correct.
Response: We apologize for this mistake. We had samples from 10 MPS patients listed in the Table 1. However for two of these patients we had only frozen brain tissues. We therefore decided to focus our analysis on the patients with both fixed and frozen tissues, but did not update the table and the text. This has now been corrected.
Line 80: Table 1 is mentioned but is missing in the manuscript and it would be very useful for a better understanding of the analysed samples!
Response: We apologise for this inconvenience. The Table 1 has been uploaded separately and had not been included into the combined PDF file sent to the reviewers. We now included the table in both text and PDF files.
In Figure 1A, in the graphic of the control brains, please change the range of the Y-axis from 0-20 to 0-15 to allow a more direct comparison with the MPS brains chart.
Response: The figure has been modified as suggested by the reviewer.
In Figure 2C, 3 and 4 the MPSII HBCB1801OC sample is missing! Please add all the data also for it or specify in the text why it is not included!
Response: In the above figures we have shown representative images of one control and 7 MPS patients. Unfortunately the quality of the fixed brain tissues of the MPS II patient HBCB1801OC was not sufficient to perform cryosectioning and conduct immunofluorescent analysis. This is now explained in the manuscript (legends to the figures 2, 3 and 4).
In Figure 2C, add a chart for GM2 and GM3 simple gangliosides, like the graph in Figure 1B. This will allow more direct visualization of the data for single MPS.
Response: The chart has been added (new Fig. 2C) and discussed (page 7, lines 178-180).
In line 185-186, review [21] is not the proper one since it doesn’t really review LSDs with impaired autophagy but the mechanisms.
In addition, several studies on LSDs and autophagy have been published in recent years. Therefore replace/add a newer and more appropriate review like Seranova et al 2017 (Essays Biochem. 2017 Dec 12;61(6):733-749).
Response: We thank the reviewer for revealing this inconsistency. The reference in question has been replaced with the one suggested by the reviewer.
In Figure 4A, the scale bar is wrong (20 µm instead of 50 µm).
Response: The scale bars are 20 µm in the panel A and 50 µm in the panel B. This is now indicated in the figure legend.
In line 245 the reference concerning MPS IIIC is missing.
Response: The reference has been added.
In the “methods” section please specify the code number of the antibodies used where possible.
Response: the catalogue numbers for the antibodies have been added to the revised section.
Reviewer 2 Report
This manuscript reports that increase of heparan sulfate (HS), decrease of keratan sulfate (KS), and upregulations of GM2, GM3, and LacCer in Mucopolysaccharidoses (MPS) patient brains. Further, the authors discovered that increases of LCB3, TNFSF10, ILB4, GFAP, and Thioflavin-S in MPS patient brains. In general, the work is interesting and important. There are, however, some concerns that require the authors’ attention before the manuscript can be accepted.
Table 1 and Supplementary Materials could not be found.
Definition of some abbreviations are missing (e.g. ILB4, LC3B, TNSF10, GFAP, and ERT).
Value/number for GD1b amount (mol %) in MPS patients would be helpful.
Page 6, line 159: Figure 3C would be Figure 2C.
Figure 3: Intensities of DAPI are dissimilar. It would be convincing if the authors could show similar DAPI level in each figure.
Author Response
The point-to-point response to the reviewer’s comments
Reviewer 2
This manuscript reports that increase of heparan sulfate (HS), decrease of keratan sulfate (KS), and upregulations of GM2, GM3, and LacCer in Mucopolysaccharidoses (MPS) patient brains. Further, the authors discovered that increases of LCB3, TNFSF10, ILB4, GFAP, and Thioflavin-S in MPS patient brains. In general, the work is interesting and important. There are, however, some concerns that require the authors’ attention before the manuscript can be accepted.
Table 1 and Supplementary Materials could not be found.
Response: We apologise for this mistake. The Table 1 has been uploaded separately and had not been included into the combined PDF file sent to the reviewers. We now included the table in both text and PDF files.
The manuscript does not have supplementary materials. The line 409 that mentioned Supplementary materials has been left from the template Word file and now is removed.
Definition of some abbreviations are missing (e.g. ILB4, LC3B, TNSF10, GFAP, and ERT).
Response: All abbreviations are now defined in the text and at the end of the paper.
Value/number for GD1b amount (mol %) in MPS patients would be helpful.
Unfortunately our data do not provide a possibility to calculate a mol % for individual gangliosides. Besides, we believe that providing this value for GD1b only would not benefit the readers. The figure 2B demonstrates that levels of GD1b are not significantly changed in MPS patients as compared with controls.
Page 6, line 159: Figure 3C would be Figure 2C.
Response: This mistake has been corrected.
Figure 3: Intensities of DAPI are dissimilar. It would be convincing if the authors could show similar DAPI level in each figure.
Response: We apologise for not-optimal quality of the images, which is related to long-term storage of post-mortem tissues in formalin. This resulted in increased autofluorescence of tissues and also affected their staining with DAPI. All images have been taken with the same settings for the confocal microscope and we do not feel that normalizing intensity of the blue channel would be appropriate.